# The Delayed Turnover of Proteasome Processing of Myocilin upon Dexamethasone Stimulation Introduces the Profiling of Trabecular Meshwork Cells’ Ubiquitylome

**DOI:** 10.3390/ijms251810017

**Published:** 2024-09-17

**Authors:** Grazia Raffaella Tundo, Dario Cavaterra, Irene Pandino, Gabriele Antonio Zingale, Sara Giammaria, Alessandra Boccaccini, Manuele Michelessi, Gloria Roberti, Lucia Tanga, Carmela Carnevale, Michele Figus, Giuseppe Grasso, Massimo Coletta, Alessio Bocedi, Francesco Oddone, Diego Sbardella

**Affiliations:** 1Department of Clinical Sciences and Translational Medicine, University of Tor Vergata, 00133 Rome, Italy; 2Department of Chemical Sciences and Technologies, University of Tor Vergata, 00133 Rome, Italybcdlss01@uniroma2.it (A.B.); 3IRCCS-Fondazione Bietti, 00168 Rome, Italygloria.roberti@fondazionebietti.it (G.R.); massimiliano.coletta@fondazionebietti.it (M.C.);; 4Department of Surgical, Medical, Molecular Pathology and Critical Care Medicine, University of Pisa, 56124 Pisa, Italy; michele.figus@unipi.it; 5Department of Chemical Sciences, University of Catania, 95125 Catania, Italy; grassog@unict.it

**Keywords:** myocilin, ubiquitin, Trabecular Meshwork Cells, E3-ligase, diGLY proteomics

## Abstract

Glaucoma is chronic optic neuropathy whose pathogenesis has been associated with the altered metabolism of Trabecular Meshwork Cells, which is a cell type involved in the synthesis and remodeling of the trabecular meshwork, the main drainage pathway of the aqueous humor. Starting from previous findings supporting altered ubiquitin signaling, in this study, we investigated the ubiquitin-mediated turnover of myocilin (MYOC/TIGR gene), which is a glycoprotein with a recognized role in glaucoma pathogenesis, in a human Trabecular Meshwork strain cultivated in vitro in the presence of dexamethasone. This is a validated experimental model of steroid-induced glaucoma, and myocilin upregulation by glucocorticoids is a phenotypic marker of Trabecular Meshwork strains. Western blotting and native-gel electrophoresis first uncovered that, in the presence of dexamethasone, myocilin turnover by proteasome particles was slower than in the absence of the drug. Thereafter, co-immunoprecipitation, RT-PCR and gene-silencing studies identified STUB1/CHIP as a candidate E3-ligase of myocilin. In this regard, dexamethasone treatment was found to downregulate STUB1/CHIP levels by likely promoting its proteasome-mediated turnover. Hence, to strengthen the working hypothesis about global alterations of ubiquitin-signaling, the first profiling of TMCs ubiquitylome, in the presence and absence of dexamethasone, was here undertaken by diGLY proteomics. Application of this workflow effectively highlighted a robust dysregulation of key pathways (e.g., phospholipid signaling, β-catenin, cell cycle regulation) in dexamethasone-treated Trabecular Meshwork Cells, providing an ubiquitin-centered perspective around the effect of glucocorticoids on metabolism and glaucoma pathogenesis.

## 1. Introduction

Glaucoma identifies a group of neurodegenerative diseases characterized by the loss of retinal ganglion cells (RGCs) and their axons, and the subsequent visual field impairment [1]. Although only 3.5% of the population over the age of 40 is affected, glaucoma is the first cause of irreversible blindness worldwide, and it has been estimated that approximatively 112 million people will be diagnosed by 2040 [2,3]. 

The main risk factors for development and progression of glaucoma are (*a*) ageing, (*b*) increased intraocular pressure (IOP), (*c*) ethnicity, and (*d*) positive family history [4,5]. However, the only therapy available is still IOP reduction by pharmacological or surgical treatments.

A progressive and robust IOP increase is observed in steroid-induced glaucoma, a severe clinical form of the disease which is developed by a subset of subjects treated with glucocorticoid (GC) therapy [6,7]. 

Steroid-induced glaucoma, which commonly affects young individuals, is asymptomatic until the late stages and it may easily lead to impaired visual function and irreversible blindness before diagnosis is carried out. Although steroid-induced glaucoma accounts for a minor fraction of clinical cases, stimulation of Trabecular Meshwork Cells (TMCs) with GCs in vitro is often adopted as an experimental model to investigate the global metabolism of this cell type and the molecular mechanisms underscoring glaucoma onset [7]. In fact, TMCs serve key roles in regulating the composition and turnover of the trabecular meshwork (TM), which is the main outflow pathway of aqueous humor (AH), a fluid which shapes the globe of the eye and nourishes the tissues of the anterior chamber [8,9]. The balance between AH production by the ciliary body and its drainage through the TM are the major determinants of IOP regulation. 

The pathogenesis of steroid-induced glaucoma, and of other clinical forms, had been long linked to the excessive synthesis, secretion, and thereby TM deposition, of myocilin (TIGR/MYOC gene) [10,11]. In fact, TMCs are the only cell type of the human body that synthesize and secrete huge amounts of this constitutively secreted glycoprotein during the secondary GC response [12,13,14,15]. 

Regrettably, the mechanisms underscoring TMC dysregulation in glaucoma are largely uncharacterized, as yet. Currently, great attention is paid to autophagy, which is a major intracellular proteolytic pathway through which unwanted and damaged proteins and organelles are delivered to lysosomes for degradation upon engulfment into a vesicle, called autophagosome [16,17,18,19,20].

In this regard, we reported altered autophagosome biogenesis through accelerated ubiquitin (Ub)-mediated turnover of key autophagy genes (Ulk1 and Atg101) in a primary human TMC strain challenged in vitro with dexamethasone (dexa), a clinically relevant GC [21]. 

In accordance with the original description of the Ubiquitin Proteasome System (UPS), ubiquitylation is the reaction catalyzed by the E1-E2-E3 enzymes that conjugate the C-terminus of ubiquitin (Ub) to the side chain of a lysine residue (K) of the protein substrate, through an isopeptide bond [22]. 

Thereafter, the Ub-tagged substrate is recognized and degraded by the 26S proteasome, a multi-catalytic assembly composed, in its canonical configuration, by one 20S core particle (CP) and one or two 19S regulatory particles (RPs) [23,24]. The RP is a multi-subunit complex which couples the recognition of the Ub-tagged substrate with its ATP-dependent unfolding and translocation into the 20S. This last particle is a hollow-barrel cylinder made up of two outer heptameric α-rings (subunits α1–α7) and two heptameric β-inner rings (subunits β1–β7), which house the three proteolytic activities, namely, chymotrypsin-, trypsin- and caspase-like [23,25,26]. 

With these premises, in this study, we first report that dexa stimulation induced a slower proteasome-mediated turnover of myocilin in a primary human TMC strain. By applying co-immunoprecipitation and gene-silencing strategies, we then propose STUB1/CHIP as a candidate E3-ligase of myocilin, further highlighting the fact that this enzyme is downregulated by dexa treatment, potentially through a proteasome-mediated turnover, as well. 

These findings have then inspired the undertaking of the first profiling of TMC’s ubiquitylome by diGLY proteomics, casting light on the main pathways regulated by Ub in resting and dexa-treated TMCs. 

## 2. Results 

### 2.1. Myocilin Does Not Accumulate in TMCs Stimulated with Dexamethasone and Epoxomicin

In a recently published paper, a faster UPS-mediated turnover of two autophagy proteins (i.e., Atg101 and Ulk1) was documented in a primary human TMC strain stimulated with dexa [21].

During a pilot study, the glycoprotein myocilin, which is a phenotypic marker of TMCs and is involved in the pathogenesis of glaucoma, was observed not to accumulate in whole-cell lysates of TMCs stimulated with dexa and epox, the most potent and selective proteasome inhibitor. Of note, myocilin was reported to be a UPS substrate by other authors, but in a different cell model and in the absence of dexa stimulation [27]. 

Hence, to investigate further the intracellular processing of myocilin in our experimental model, TMCs were stimulated with 500 ng/mL dexa or the equivalent volume of ethanol (i.e., the solvent of dexa) for 4 days in vitro. 

At day 4 of the treatment, 1 µM epox, or epox solvent (DMSO) DMSO (i.e., the solvent of epox) as control, was delivered 3 h before cell harvesting and whole-cell lysates were prepared for Wb analysis [28]. This short incubation time is mandatory to rule out off-targets effects of proteasome inhibition. Filters were then probed with an anti-myocilin antibody, and with an anti-tubulin antibody as loading control (Figure 1). 

First, a robust increase (~10-fold) of myocilin staining in dexa-treated cells with respect to vehicle-treated cells was documented (both labeled as untreated in Figure 1), confirming the TMC strain identity and the effectiveness of dexa stimulation. However, in the presence of epox, whilst vehicle cells showed a significant increase in myocilin staining (~2-fold change with respect to levels of untreated cells), no further increase in myocilin intensity was observed in dexa-treated cells, compared to the absence of the proteasome inhibitor (untreated cells) (Figure 1). 

To delineate further this finding, also considering that myocilin is a constitutively secreted glycoprotein, TMCs were stimulated with 20 µM Brefeldin A (BrefA), which inhibits protein secretion through the Golgi pathway, or 20 µM chloroquine (CQ), which inhibits lysosomal acidification, causing accumulation of autophagy substrates [29]. Both these two drugs were administered 2 h before cell harvesting. Also, in this case, the short incubation time is required to rule out off-target effects. 

Compared to untreated cells, BrefA and CQ turned out to be effective in increasing myocilin immunostaining in both vehicle- and dexa-treated cells (Figure 1). 

### 2.2. Defective Myocilin Clearance by Proteasome in Dexa TMCs

To strengthen the hypothesis of a defective proteasome-mediated clearance of myocilin in dexa-treated cells, a native-gel study was set up. This approach applies to the separation of intact proteasome assemblies (3000–750 kDa) under non-denaturing conditions (Figure 2) [30]. 

In our previous study, global activity, composition and abundance of proteasome particles was found to be fully comparable between these experimental conditions in the same TMC strain [21].

Therefore, in this study, intact proteasome assemblies were again separated, transferred to a nitrocellulose filter and probed with antibodies raised against PSMA3 (i.e., the α4 subunit of the 20S), myocilin and Ub (Figure 2, IB panel). 

Immunostaining of the proteasome subunit highlighted no obvious differences in the relative abundance (and capped/uncapped particle ratio) of the three main particles across all the experimental conditions, and was used as loading control. The intensity of myocilin was robustly immunodetected in the correspondence of the capped proteasome assemblies (in particular, the 30S) of vehicle-treated cells, and its intensity strongly increased in the presence of epox under this experimental condition. This behavior is consistent with that of substrates undergoing proteasomal processing (Figure 2, upper histogram).

Conversely, in the presence of dexa (and the absence of epox), myocilin intensity in the correspondence of the 30S was lower than that documented in vehicle-treated cells (and in the absence of epox). Moreover, epox delivery induced only a slight, and not significant, increase in myocilin content in association with the proteasome particle, compared to the absence of epox, in dexa-treated cells (Figure 2, upper histogram). 

As internal control, proteasome particles were further probed with an anti-Ub antibody (Figure 2 and lower histogram). 

Although crisp Ub-positive bands could not be obtained, likely because of the intrinsic nature of Ub-proteins and their co-migration with proteasome particles, the Ub-positive intensity was uniformly distributed across the capped assembly’s region of interest. The intensity of Ub staining turned out to be slightly decreased in dexa- vs. vehicle-treated cells in the absence of epox. However, in this case, delivery of the proteasome inhibitor induced a robust increase in Ub immunostaining in both vehicle- and dexa-treated cells compared to their own control (i.e., the absence of epoxomicin). 

As a whole, recalling that native-gel studies offer a qualitative interpretation of data, these data suggested that myocilin was poorly degraded by proteasome particles in the presence of dexa stimulation. 

### 2.3. Identification of STUB1/CHIP as the Putative E3-Ligase of Myocilin through a Multidisciplinary Approach

Considering the already cited data regarding altered ubiquitin signaling and autophagy dynamics, and the conceptual organization of the UPS, we reasoned that the most likely explanation for this phenomenon would have been the downregulation of the E3-ligase-targeting myocilin, which, to our knowledge, has not been characterized yet. Therefore, a research plan was undertaken to identify the putative myocilin/E3 ligase pair in TMCs.

First, the Ubi-browser software 2.0 was used to predict possible protein:E3-ligase pairs based on domains and structural features [31]. The search retrieved two candidate E3-ligases by searching under default conditions: STUB1/CHIP and SYVN1. 

Thereafter, a co-immunoprecipitation (co-IP) assay was set up to verify the possible interaction of myocilin with either one of the two enzymes, starting from cell pellets of untreated TMCs. Based on the findings here discussed, a co-IP studies using dexa-treated cells would have provided less chance of identifying the candidate E3-ligase.

Thus, proteins of cell lysates were equally divided and addressed toward co-IP with the anti-myocilin antibody or species-matching control IgGs. 

An aliquot of cell lysate (referred to as input) together with the myocilin- and IgG co-immunoprecipitated samples were then separated by denaturing and reducing Wb (Figure 3). Filters were probed with the anti-myocilin (rabbit), anti-STUB1/CHIP and anti-SYVN1 antibodies. 

Myocilin immunostaining confirmed a successful pull-down assay. The doublet band of the glycoprotein (55–57 kDa), and also its natural N-terminal fragment, were detectable only in the input and in the myocilin co-IP lanes, but not in the IgG co-IP lane (Figure 3, left panel). The anti-STUB1 antibody revealed the presence of a band with the predicted molecular weight (32 kDa) in the myocilin co-IP fraction (and indeed, input), but not in the control IgG fraction (Figure 3, middle panel). Conversely, the anti-SYVN1 antibody detected the protein (89 kDa) exclusively in the input fraction (Figure 3, right panel). 

To further validate the possibility that STUB1/CHIP is an E3-ligase of myocilin, a gene-silencing strategy was then set-up. 

TMCs were then grown for 72 h in the presence of three different anti-sense oligonucleotides (27-mer siRNA) targeting STUB1/CHIP (defined #A, #B, #C) or in the presence of a non-targeting siRNA (referred to as Pool), to rule out off-target effects. As internal control, cells were further grown in the presence of the siRNA vehicle (Ctrl).

Whole-cell lysates were then harvested and analyzed by Wb (Figure 4). STUB1/CHIP silencing was successfully reached, using two out of the three siRNAs delivered (#B and #C, but not #A). As expected, the siRNA-Pool did not affect STUB1/CHIP intensity with respect to untreated cells. In this case, GAPDH immunodetection was used as internal control (Figure 4). 

In the presence of #B and #C siRNAs, myocilin content significantly increased compared to all other conditions tested, confirming that downregulation of STUB1/CHIP was associated with myocilin accumulation in TMCs. Since the glycoprotein is constitutively secreted, to circumstantiate further this finding a Wb analysis of cell culture supernatants (harvested after 72 h of incubation) was carried out. Interestingly, myocilin intensity was robustly increased in the culture medium in the presence of #B and #C siRNAs, with respect to all other conditions tested. 

Finally, to rule out the possibility that myocilin accumulation was a consequence of ER stress induced by STUB1/CHIP silencing, filters were further probed for a ER stress marker such as calnexin. 

### 2.4. STUB1/CHIP Is Downregulated by Dexamethasone, Likely through Altered Turnover

On the basis of the data discussed so far, protein and transcript levels of STUB1/CHIP were assayed in vehicle- and dexa-treated TMCs (for 4 days) by Wb and RT-PCR (Figure 5).

The protein content of STUB1/CHIP was significantly affected by dexa: an evident decrease was documented in dexa-treated cells compared to vehicle-treated cells (Figure 5A). Surprisingly, the transcript levels of STUB1/CHIP (but also of SYVN1, checked as control) were comparable between vehicle- and dexa-treated cells (Figure 5B), suggesting that the decrease in STUB1/CHIP content was not caused by a transcriptional downregulation of its gene. As a further control, myocilin transcripts were assayed in parallel, confirming that transcription of the MYOC/TIGR gene was robustly triggered by dexa treatment (Figure 5B).

Interestingly, when whole-cell lysates were prepared from TMCs stimulated with dexa (or from vehicle) in combination with epox (see Figure 1), an evident increase in STUB1/CHIP band intensity was observed in dexa-stimulated cells in the presence of the proteasome inhibitor, compared to its absence (Figure 5A). Conversely, epox treatment had no obvious effects on STUB1/CHIP levels in vehicle-treated cells (Figure 5A). 

To further circumstantiate this finding, the content of two relevant Heat Shock Proteins (HSPs), HSP90 and HSP70, which have been report to interact and assist STUB1/CHIP activity, were assayed [32,33].

In the absence of epox, dexa treatment did not induce any obvious effect on the basal level of these HSPs, compared to vehicle-stimulated cells. Delivery of the proteasome inhibitor turned out to stimulate a parallel accumulation of HSP70 in both vehicle- and dexa-treated TMCs, compared to the basal levels of the protein for the two experimental conditions. Conversely, epox did not induce accumulation of HSP90, either in vehicle- or dexa-treated cells.

### 2.5. Profiling of the TMC Ubiquitylome by diGLY Proteomics

The whole set of data confirmed the working hypothesis that signaling through Ub conjugation was significantly affected by dexa treatment in the primary human TMC strain. 

Therefore, a diGLY proteomics workflow was set up to figure out the ubiquitylome of TMCs under resting conditions (vehicle-treated cells) and under dexa stimulation. 

The LC/MS analysis identified and quantified a remarkable 2330 unique Ub-remnant by setting a filter for peptides with q ≤ 0.01 in the output file. 

By further filtering for Ub-remnants identified in at least *n* = 2 samples of the same experimental group, a considerable overlapping (1250 unique peptides) between the ubiquitylome of vehicle- and dexa-treated cells was documented (Figure 6A). Encouragingly, the coefficient of variation (CV) was <25% for more than 50% of peptides, and displayed a very similar distribution between dexa- and vehicle-treated cells (Figure 6B). Moreover, an average 2.11 and 2.18 Ub-sites for protein were identified in dexa- and vehicle-treated cells, respectively. Conversely, 254 and 347 Ub-peptides were classified as potentially specific for dexa- and vehicle-treated cells, respectively. Ub-peptides were then analyzed for fold-change (FC) intensity.

By applying a Log_2_ FC ± 0.57 and a *p* ≤ 0.05 adjusted by Benjamini–Hochberg correction, a remarkable number of Ub-remnants turned out to be upregulated or downregulated in dexa- vs. vehicle-treated TMCs (Figure 6C). The whole list of Ub-proteins found as downregulated or upregulated (in the dexa/vehicle ratio), along with the Ub-remnant identified, is provided in Table 1 and Table 2, respectively.

To cluster and rationalize data, the proteins from which the Ub-peptides were reported, were inputted to DAVID software (v. 6.8), and Gene Ontology enrichment charts, including Biological Processes (BPs), Molecular Function (MF) and Cellular Components (CCs) were generated. In all cases, *p* ≤ 0.05 adjusted by Benjamini–Hochberg correction was set as the threshold for statistical significance of enriched terms.

Regarding vehicle-treated cells, among those showing the highest gene ratio in the CC chart, terms of cell polarity and terms referable to tight junctions and protein kinase complexes were identified (Figure 7A). In this regard, three individual Ub-sites were identified for Gap junction alpha-1 protein (P17302). Thereafter, clusters belonging to lipid-mediated structures, such as membranes rafts and microdomain, as well as early endosome membranes, were identified as enriched (Figure 7A).

Whilst the MF enrichment retrieved no enriched terms, the BP processes of vehicle-treated cells introduced interesting findings, including the following: regulation of wound healing, processes related to immune-system regulation, gliogenesis, vesicle-mediated transport to the plasma membrane and, remarkably, mechanisms of cell-cycle control, with a particular focus on the G1/S transition (Figure 7B). With regard to this point, although the enrichment chart displayed a relative low count for this term, it is worth pointing out that four known Ub-sites (PhosphositePlus v6.7.1.1) of G1/S-specific cyclin-D1 (P24385) were found to drop in dexa-treated cells, together with known Ub-sites of Cell-cycle control protein 50A (Q9NV96) and of cyclin-dependent kinase inhibitor (P38936).

Regarding the dexa-treated cells, a very different pattern was observed. In fact, the CC chart highlighted a strong enrichment in terms identifying cellular compartments and cell polarity, differently from those identified in vehicle-treated cells. In this case, enrichment in focal adhesion and cell-substrate junction, together with the baso-lateral portion and sarcolemma were further enriched (Figure 7C).

Moreover, myofibrils, contractile fibers and lysosome, lytic vacuolar membranes, membrane microdomain and lipid rafts terms were enriched, as well. With regard to these last points, it is worth pointing out that a known Ub-site of phospholipid phosphatase 3 (O14495) showed a log_2_FC ≥ 2.

In the case of dexa-treated cells, the MF chart found terms significantly enriched including specific mechanisms of cell adhesiveness (e.g., cadherin bindings), and mechanisms of Ub-conjugation and UPS-mediated turnover and of GTPase binging (Figure 7D).

In accordance with terms identified in the two previous charts, the BP chart highlighted a long list of terms which referred to processes very relevant for the TMC metabolism and the pathological transformation they undergo during dexa-stimulation.

The most robustly upregulated terms included mechanisms of connective tissue remodeling, cell adhesiveness, and growth control. With regard to connective tissue remodeling, the relevant proteins showing upregulated Ub-sites were laminin-subunit β2 (P55268), integrin α-5 (P08648), and vimentin (P08670), which showed up with three individual Ub-sites, prolyl 4-hydroxylase subunit α-2 (O15460) and procollagen-lysine, 2-oxoglutarate 5-dioxygenase 2 (O00469).

In accordance with the working hypothesis which has inspired the undertaking of the diGLY workflow, terms referable to the regulation of targeted proteolysis by the UPS were found to be enriched in dexa-treated cells.

In this case, relevant proteins characterized by increased ubiquitination at known sites in the presence of the drug were caveolin-1 (Q03135), Rho-related BTB domain-containing protein 3 (O94955), F-box only protein 32 (Q969P5), also known as atrogin-1, catenin β1 (P35222), and E3 ubiquitin-protein ligase Itchy homolog (Q96J02).

Strikingly, mechanisms of sensory- and visual-system development were enriched with several unique terms (e.g., visual system development, eye development, camera-type eye development (Figure 7E)).

## 3. Discussion

After having reported impaired autophagosome biogenesis as a consequence of the faster turnover of Atg genes, (i.e., Atg101 and Ulk1) [21], a delayed proteasomal processing of intracellular myocilin was here identified in a primary human TMC strain stimulated with dexamethasone (see Figure 8, for a schematic summary of the main findings). This finding was bona fide, and associated with a STUB1/CHIP dysregulation, proposed as a putative E3-ligase of the glycoprotein, and probably not the only one, as promiscuity between E3-ligase:substrate pairs exist [11,13,27,34]. Remarkably, these datasets have encouraged us to run the first profiling of the TMC ubiquitylome by diGLY proteomics, thereby introducing novel perspectives on the main biological processes, molecular functions and cellular components primarily regulated by this post-synthetic modification in resting and dexa-treated TMCs.

Before introducing any further consideration, we recognize that that this study is burdened by the intrinsic limitation of having been run on a single TMC strain. However, additional commercial strains failed the proof of identity (i.e., myocilin induction by GC and morphology in vitro), a finding already described by other authors, and isolation of primary TMCs is not allowed by current local ethical laws [35]. However, we feel that, without claiming that our findings have a universal application to TMCs in vitro and in vivo, the wide repertoire of methodological approaches here adopted introduces a set of new findings in the field, which can stimulate further studies.

With respect to the myocilin behavior, the data here reported suggest that its delayed turnover in TMCs stimulated with dexamethasone is caused by its impaired ubiquitylation. In fact, unlike vehicle-treated cells, proteasome inhibition by epox (i.e., a potent and selective proteasome inhibitor), did not induce significant accumulation of intracellular myocilin nor increase the association of the glycoprotein with intact proteasome assemblies, a behavior expected instead for a proteasomal substrate, and actually documented for Ub-proteins in both vehicle- and dexa-treated cells. Nevertheless, proteasome activity and composition under these experimental conditions were found to be unaltered in this TMC strain [21]. Moreover, CQ and BrefA induced accumulation of intracellular myocilin, either in the presence of dexa or not, thereby conferring some specificity to the epox effect, and envisaging that dexa treatment does not significantly alter the sorting and trafficking pathways through which myocilin is physiologically routed.

Moreover, bioinformatic and molecular approaches (i.e., co-immunoprecipitation and gene-silencing strategies) indicate that the delayed myocilin turnover is seemingly caused by the dexa-mediated downregulation of STUB1/CHIP, which is then proposed as a candidate E3-ligase of this glycoprotein.

Interestingly, STUB1/CHIP is a main E3-ligase that surveys the protein homeostasis inside the endoplasmic reticulum (ER), thus playing a central role within the dynamics of the Unfolded Protein Response (UPR).

The ER is the organelle inside which myocilin is translocated towards the Golgi network for extracellular secretion; this strengthening, which is at least based on subcellular compartmentalization, assumes the possibility that STUB1/CHIP actually ubiquitylates myocilin [35]. The behavior observed for STUB1/CHIP seems to support the hypothesis of an altered Ub-signaling in dexa-treated TMCs, since the marked drop of STUB1/CHIP levels in the presence of the drug did not rely on transcriptional downregulation, but, probably, on its altered turnover, even though a clear mechanism for interpreting the STUB1/CHIP drop in dexa-treated cells is far from being understood. In this regard, the parallel analysis of HSP70 and HSP90, which are two relevant HSPs reported to interact with STUB1/CHIP, assisting its biological function, may confer some specificity to the STUB1/CHIP finding [32,33]. In fact, whilst dexa treatment did not alter the basal content of the two HSPs, epox delivery induced accumulation of HSP70, but not HSP90, through patterns comparable between vehicle- and dexa-treated TMCs. At this stage, we cannot exclude the fact that HSP70 accumulation reflects its transcriptional upregulation by epoxomicin proteotoxicity.

In general, the STUB1/CHIP finding may not be so unexpected, since E3-ligases are subjected to Ub-mediated turnover, like any other protein of the cell. On the other hand, a marked lowering of STUB1/CHIP levels has already been reported for aberrant autophagic fluxes in neurodegenerative disease [36], being associated with a parallel downregulation of Ser757 phosphorylated Ulk1. Therefore, a link can be envisaged between these data and those previously reported by us in dexa-treated TM [21].

Then, to figure out global mechanisms of altered Ub-signaling, the first characterization of TMC ubiquitylome was here undertaken by diGLY proteomics, a workflow developed to profile the Ub-code of biological samples, and which allows for the uncovering of key pathophysiological mechanisms regulated by this post-synthetic modification [37,38,39].

Taking into consideration the technical limitations (discussed in the Methods Section), the number of Ub-remnants identified and quantified, though satisfactorily, cannot represent the whole repertoire of Ub-proteins, but those which are most abundant. For this reason, a recently developed DIA-method based on neural networks implemented in the DIA-NN software (v. 1.9), which was set up and validated for diGLY proteomics (but not only for this), was used to maximize the depth and coverage of Ub-remnant identification [40,41].

Based on the number of targets identified, it is not surprising that Ub-peptides of myocilin were not identified. In facts, in accordance with the working hypothesis, in dexa-treated cells the fraction of Ub-myocilin in the intracellular space should be very low, and it is worth recalling that the glycoprotein is constitutively secreted.

However, the rationalization of diGLY datasets strongly confirms the fact that dexa actually induces a profound re-arrangement of TMCs ubiquitylome, this being inherently supported by further enrichment in Ub-remnants and related GO terms identifying the mechanisms of Ub-conjugation and the proteasome-mediated digestion of substrates in this experimental group. Specifically, within the pool of Ub-remnants enriched in dexa-treated TMCs, known ubiquitylation sites of proteins and E3-ligases that play a pivotal role in cell metabolism and signaling pathways, such as caveolin-1, atrogin-1 and, remarkably, β-catenin, were identified. With respect to this last point, β-catenin is widely recognized as serving crucial roles for Wnt (Wnt/β-catenin) signaling, which is a highly conserved pathway regulating key cellular functions such as cell proliferation, differentiation, migration, and apoptosis [42]. Nevertheless, this pathway is thought to be central for TMCs metabolism and the regulation of IOP, and to further represent a candidate for glaucoma therapeutic strategies.

Furthermore, all terms identified underscore the involvement of Ub signaling in pathways strongly associated with the alteration TMCs develop in steroid-induced glaucoma. In this regard, terms referring to subcellular compartments involved in cell adhesiveness and lipid metabolism, with phospholipid phosphatase 3 emerging as the protein with the highest log_2_FC, motility and secretion polarity support a conceptual correlation with the dysregulation of the TM typical of glaucoma pathogenesis and, in particular, the stiffness of the TM, which impairs the AH drainage.

Additionally, it is worth underscoring the fact that Ub-peptides of proteins involved in cell proliferation and cell-cycle control were identified as downregulated in dexa-treated cells. This finding may have implications in interpreting the reduced proliferation and lower TMC count and density documented in animal disease models [3,8,9]. Moreover, this finding is consistent with the upregulation of the p21 protein, which is a master regulator of the G1/S transition, and of cyclin D, under the same experimental condition, previously reported by us [21,43].

TMC dysregulation, which develops during glaucoma pathogenesis, is a matter of intense debate across the scientific community. Different metabolic and signaling pathways have been recently uncovered, which are altered in the presence of stimuli, and which are classified as risk factors for disease onset, such as aging, redox unbalance, glucocorticoid stimulation, and the expression of mutated proteins (including myocilin) [3,4,44,45].

However, the molecular mechanisms underscoring these alterations have been poorly characterized, as yet.

This limitation is mirrored by the absence of specific therapeutic approaches other than reducing the impact of risk factors on disease progression, such as lowering the IOP by surgical or pharmacological approaches, which is still the one proposed for subject with the disease.

A thorough understanding of biological and patho-physiological processes can be achieved by figuring out the factors involved, in most cases proteins, and, often, how they are modified, either on the basis of inherited or acquired conditions. The post-translational modifications of proteins (PTMs), which is, in general, a hallmark of the activation/repression of signaling pathways, is a poorly explored topic in glaucoma pathogenesis [46,47].

Among the most abundant PTMs, ubiquitylation is gaining considerable relevance in deciphering the dynamics of the pathogenesis of several diseases, spanning neurodegeneration to cancer [22,48]. We would underline the fact that a tryptic Ub-remnant is generated not only upon the trypsin cleavage of Ub, but also of ISG and NEDD, which identify PTMs other than ubiquitylation, referred to as ISGylation and NEDDylation, respectively. However, they normally account for <5% of Ub-remnants. Therefore, even if some of the Ub-remnants identified may refer to these modifications, they are likely a very minor fraction, and are unlikely to impact data interpretation.

Importantly, there is now compelling evidence for the fact that differential ubiquitylation has a great impact on the biological properties of proteins, not only for clearance by the proteasome, as originally hypothesized.

The huge heterogeneity of conformations through which any given substrate can be potentially decorated with Ub poses the challenge of uncovering the precise biological roles of any specific Ub-linkage and topology [22]. This consideration gains even more relevance assuming that each Ub configuration may have a specific role in disease pathogenesis and may represent a potential neoepitope, as documented for Ub-tau (MAPT gene) species in tauopathies [49].

Remarkably, the Ub-code, as well as its disease-related modification, is primarily regulated through differential expression of E3-ligases, which are the enzymes conferring specificity to the substrate selected for Ub-tagging.

Considering that the human genome encodes for >600 E3-ligases, and thousands of proteins are routinely ubiquitylated, it seems clear that each E3-ligase has specificity for dozens of substrates, and that downregulation or upregulation of just one enzyme can bring about unpredictable alterations of cell metabolism by increasing or decreasing the turnover of its substrates [50,51].

The interaction of E3-ligases with their substrates often occurs transiently, and with low affinity. Therefore, to screen-out the vast repertoire of enzyme:protein combinations is challenging, but it represents a field of research worth being explored, as witnessed by the introduction of several drugs targeting E3-ligases in human therapies (PROTACs) [52].

In conclusion, this study seems to strengthen the need to explorie the Ub-signaling pathways in glaucoma onset and progression, in order to clarify its pathogenic role, but also to envision possible future approaches based on Ub-conjugation which stimulates rescue.

As a matter of fact, the evidence here reported may further suggest that the well-known increase in myocilin in TMCs originates not only from the transcriptional upregulation of the MYOC/TIGR gene, which has been confirmed to be a predominant mechanism also, by us, but that it is also sustained by a reduced proteasomal turnover of the glycoprotein.

## 4. Materials and Methods

### 4.1. Cell Culture

The human primary TMC strain was purchased from Cell Application (San Diego, CA, USA) and grown in DMEM supplemented with 10% FBS plus supplements (antibiotics and non-essential amino acids) under standard aerated conditions (37 °C, 5% CO_2_) Sigma-Aldrich (St. Louis, CO, USA). All experimental procedures were carried out within the 9th passage.

Validation of TMC strains’ phenotypical identity is mandatory before any experimental program is undertaken. A recognized method of probing the identity is to monitor the expression and/or intracellular content of the glycoprotein myocilin (TIGR/MYOC gene) in the presence of GC stimulation, since this cell type is the only known one of the human organisms which triggers myocilin transcription as a secondary GC response [28].

This check was preliminarily carried out, and the data presented throughout the manuscript support the TMC identity of our strain.

Dexamethasone (Dexa), Epoxomicin (Epox), Chloroquine (CQ) and Brefeldin A (BrefA), were purchased from Sigma-Aldrich (St. Louis, CO, USA) and solubilized according to the manufacturer’s instructions.

As control, cells were stimulated with the equivalent volume of dexa solvent [21].

### 4.2. Native-Gel Electrophoresis

Crude cell extracts (e.g., soluble fraction of the cell containing proteins) were isolated under non-denaturing condition by freeze–thawing cycles, using an osmotic buffer (250 mM sucrose, 20% glycerol, 25 mM Tris-HCl, 5 mM MgCl_2_, 1 mM EDTA, 1 mM dithiothreitol [DTT], 2 mM ATP, pH 7.5) [30,53,54]. Protein extracts were then cleared by centrifugation at 13,000 rpm for 20 min at 4 °C, and protein concentration was determined by Bradford assay.

For each experimental condition, 30 µg of total protein was separated under native conditions, through a 3.5% acrylamide gel.

Separated particles were denatured by incubating the gel in 5% SDS for 10 min and were then transferred to a nitrocellulose filter. Unspecific binding sites were blocked using a 5% fat-free milk solution solubilized in phosphate-buffered saline (PBS) containing 0.1% Tween-20 (Sigma Aldrich, St. Louis, CO, USA).

Particles were then probed with antibodies raised against the following proteins: α4 (i.e., 20S subunit), myocilin and ubiquitin.

The anti-myocilin antibodies used for this immunostaining (but not only for this) were purchased by Sigma-Aldrich (clones 4F8 and 7.1) (St. Louis, CO, USA). Although the antibodies appear to work similarly, the first one (used for the data for Figure 1) had an apparent higher affinity for myocilin staining and, in particular, for the upper MW species of the myocilin doublet band, as well as for high-molecular-weight species of unknown identity.

The other antibodies used were all supplied by Protein-tech Group (Manchester, UK).

All antibodies were diluted 1:3000 in 0.1% Tween-PBS fat-free milk and then incubated with a Horseradish Peroxidase-conjugated anti-rabbit or anti-mouse IgG antibody (Biorad, Hercules, CA, USA), diluted 1:50,000 in 0.2% Tween-PBS fat-free milk.

### 4.3. Western Blotting

For denaturing and reducing Western blotting (Wb), TMC were lysed in standard RIPA buffer supplemented with protease/peptidase and phosphatase inhibitor cocktails (Sigma-Aldrich, St. Louis, CO, USA), and cleared by centrifugation at 13,000 rpm for 30 min, at 4 °C. Also, in this case, protein normalization was performed by Bradford assay.

Gel with a different percentage of acrylamide (10–15%) or pre-cast (4–20% acrylamide) was used, and lysates or cell-culture supernatants were separated under denaturing (Laemmli buffer) and reducing (DTT) conditions Sigma-Aldrich (St. Louis, CO, USA).

For subsequent Wb procedures, parameters already described for native-gel electrophoresis study were used.

Antibodies raised against Myocilin (rabbit), STUB1/CHIP, synoviolin (SYVN1), calnexin, and CHOP, as well as internal controls, such as GAPDH and tubulin, were purchased from Protein-tech Group (Manchester, UK). Antibodies raised against HSP90 and HSP70 were purchased from Cell Signaling Technologies (Danvers, MA, USA).

The choice of the loading control was made on the basis of the determination of the dynamic linear range for the proteins used [21]. GAPDH was not used in the presence of dexa, since this treatment was reported to alter its content. Uncropped figures are uploaded as a separate .pdf file.

### 4.4. Gene-Silencing Procedures

To silence the expression of the STUB1/CHIP gene, 3 different 27-mer small interfering RNAs (siRNA), listed #A–#C, 100 nM, targeting the mRNA sequence, were used (OriGene Technologies, Rockville, MD, USA). As specified by the manufacturer, it is expected that 1 out of the 3 oligonucleotides is ineffective in inducing the silencing of the target. In this case, that labeled as #A was found not to target the expression of STUB1/CHIP. Therefore, data obtained using this batch were interpreted as a further internal control. Cells were stimulated with a pool of non-targeting 27-mer siRNAs (100 nM) under the same experimental conditions, to rule out off-targets biological effects. In each experiment, cells were further left untreated. After 48 h of incubation, to support the viability of cells, 2% FBS was added to the culture medium.

In all cases, TMCs were seeded at 75% confluence in a 6-well plate and stimulated for 72 h, before lysis by RIPA buffers. Subsequent analyses (Wb) were carried out as described previously. Supernatants were collected after 72 h of incubation, and analyzed by Wb, as well (20 µL of supernatant per experimental condition).

### 4.5. Co-Immunoprecipitation

The anti-myocilin antibody (7.1 clone) previously described was used for this workflow. Species-matching non-specific immunoglobulin G (IgGs) were purchased from the same supplier.

M-270 epoxy Dynabeads (Thermo Scientific, Waltam, MA, USA) using an optimized version of the protocol suggested by the manufacturer (5 μg of Ab/1.5 mg of beads) were used. Magnetic beads, coated with antibodies and stored at 4 °C in PBS, 0.02% NaN_3_, were washed three times with lysis buffer, and added to the soluble fraction of cell lysates. Total-protein extracts were obtained by rinsing the cells twice with ice-cold PBS, followed by the addition of ice-cold lysis buffer (Co-IP), supplemented with 1 mM DTT, 50 mM NaCl and inhibitors, as specified above. Harvested cells were washed once in PBS, and the pellet was resuspended in 1/9 ratio of cell mass to extraction buffer, supplemented with a protease inhibitor mixture, according to manufacturer’s instruction. Cells were incubated for 15 min in ice, and centrifuged at 2000 rpm for 5 min at 4 °C. The extract was used immediately for co-immunoprecipitation. Purification was achieved by slow mixing at 4 °C.

### 4.6. Gene-Expression Analysis

RNA was isolated with Trizol reagent (Life Technologies, Carlsbad, CA, USA) from all collected samples. First-strand cDNAs were synthesized from 1 μg of total RNA in a 20 μL reaction with reverse transcriptase, according to manufacturer’s instructions (BioLine, London, UK). Real-time PCR (CFX, Biorad) was performed with 50 ng of cDNA, using a SYBR green Master Mix (Biorad, Hercules, CA, USA). In all cases, β-actin was used as internal control. All primers used in these experiments are reported in Table 3.

### 4.7. DiGLY Proteomics Workflow

TMCs were cultivated and stimulated with dexa, following the scheme and dosage indicated above, and 1 µM epoxomicin was delivered 3 h before cell harvesting. This approach is deemed to enrich the pool of endogenous Ub-proteins without introducing ubiquitylation events related to the stress induced.

Cell pellets were then lysed in 8 M urea, 150 mM NaCl, 25 mM Tris-HCl, pH 8. The lysis buffer was supplemented with phosphatase inhibitor (1 mM sodium orthovanadate, 1 mM β-glycerophosphate) and peptidases and protease inhibitors, as indicated above. The lysate was sonicated and cleared by centrifugation before determination of protein concentration by BCA assay. The TMC ubiquitylome was then resolved by diGLY proteomics. This workflow is based on the immunoaffinity enrichment of peptides decorated with Ub-remnants before analysis by liquid chromatography, coupled with mass spectrometry (LC-MS) [37,39].

The principle of the method relies upon the generation of Ub-remnants during the trypsin digestion of the sample, with no variation from a standard procedure for shot-gun/bottom-up proteomics.

In fact, Ub-remnants are peptides stretching across the two adjacent tryptic sites on the parental protein, with the two C-terminal glycine residues of Ub still anchored to the lysine (K) residue through the natural isopeptide bond. The generation of this branched peptide is made possible by the presence of a trypsin cleavage site on Ub, after the two glycine residues.

After preparation of the tryptic digest, Ub-remnants are enriched by using an anti-diGLY antibody and then analyzed by mass spectrometry (see below) (Cell Signaling Technology, Danvers, MA, USA)

For each technical replicate, approx. 250 µg of proteins were enrolled in the study. Proteins were first treated with 5 mM DTT, to reduce disulfide bridges (45 min at room temperature [r.t.])m and next alkylated with 10 mM iodoacetamide (30 min at r.t. in the dark). At the end of the alkylation step, 5 mM dithiothreitol was further added. Protein digestion was then performed in two steps: (i) the first one, with lysil-c-endopeptidase and trypsin (1:100 and 1:20 enzyme-to-protein ratio, respectively, 2 h, r.t.) and (ii) the second one, with trypsin (1:20, o.n., 25 °C), after diluting with Tris-HCl 50 mM to reach 1.5 M urea concentration.

Thereafter, the tryptic digests were cleared by centrifugation (13,000 rpm for 10 min), acidified (1% trifluoroacetic acid (TFA) and cleaned by using Hypersep C18 columns (Thermofisher scientific). The resin was first wetted with 100% acetonitrile (ACN) (0.5 mL), conditioned with 0.1% TFA (2 × 1 mL). Then, the sample was loaded, washed with 2 × 1 mL 0.1% TFA and 0.5 mL of 5% ACN-0.1% TFA, and, finally, eluted with 3 × 0.5 mL 50% ACN-0.1% TFA.

Eluted peptides were then dried down in the Centrivap Vacuum Concentrators system (Labconco Corporation, Kansas City, MO, USA), resuspended in 5% ACN, 0.1% formic acid (FA) quantified by BCA-peptide assay, and then split into different aliquots containing 150 µg peptides each.

Thereafter, the Ub-remnants were enriched using the diGLY kit, following manufacturer’s instruction, with the only variation being that the volume of buffer (calibrated for 1 mg of peptides) was adjusted to meet the low quantity of starting peptides.

At the end of the procedure, peptides were eluted in 0.15% TFA, desalted using Pierce C18 stage tips, dried, and resuspended in loading solvent (5% ACN, 0.1% FA) for MS analysis.

Samples were analyzed using an Orbitrap Exploris 240 (Thermo Fisher Scientific, Waltham, MA, USA) online, with a nano ultra-high-pressure liquid chromatography system (Dionex, Ultimate 3000).

Samples were analyzed using an 88 min LC-MS method. Mobile phase flow rate was 250 nL/min. Solvent A was 0.1% FA, whereas Solvent B was 90% ACN/0.1% FA.

The LC-MS/MS method used the following gradient profile: (min: %B) 0:2; 2:6.7; 62:34.4; 67:55.5; 72:100.

The acquisition was carried out by Data Independent Acquisition (DIA), setting the following parameters:

Orbitrap resolution during MS scan: 120.000; 85 scan events over a Scan Range (m/z) of 375/1650, with an isolation window of 15 m/z and 1 m/z overlap between consecutive windows. Window placement optimization was enabled; Normalized AGC Target (%): 300; HCD Collision Energies 30%; Orbitrap resolution for MS/MS 30000.

The TMC ubiquitylome was here explored as a biological duplicate, each one as a technical duplicate (with each sample being singly injected). Based on technical parameters (calibrated on additional experimental models and on the protocols available), this setting turned out to be a compromise between the number of identifiable Ub-peptides, the starting material that can be obtained cultivating a primary human TMC strain before senescence, and the many different experimental procedure set-ups using this strain. In this regard, it is worth pointing out that diGLY is typically run on milligrams of protein input and that, to our knowledge, the workflow has never been applied to a human primary cell line [37,55].

Furthermore, TMCs have a very big size, implying that they are dense in structural components that account for a significant mass of non-Ub proteins.

### 4.8. Bioinformatic Analysis

Raw DIA fields were inputted into DIA-NN software (v. 1.9), following a bioinformatic pipeline validated for diGLY proteomics by the authors [40]. A library free search was selected (using a human FASTA database, UniProt 9606), enabling the generation of a sample-specific DIA library for re-processing of the files during the match-between-runs (MBR).

Output files were analyzed by R-Studio. The raw-output file generated by DIA-NN is uploaded separately as an .xls file.

Upregulated and downregulated proteins, identified by their accession number, were submitted to Database for Annotation, Visualization and Integrated Discovery (DAVID, v. 6.8) and enrichment charts were computed with R-Studio (v. 4.4).

### 4.9. Statistical Analysis

Unless otherwise indicated, in all figures the values reported are the Mean ± Standard Deviation (SD) of *n* = 3 independent biological replicates, except gene-silencing studies, which were run twice (*n* = 2).

Unpaired τ *t* Student’s test, non-parametric Mann–Whitney one-way ANOVA followed by Tukey post hoc significance tests were used, depending on the experiment.

Statistical significance was attributed to differences characterized by *p* ≤ 0.05. Data elaboration and statistical analysis were performed by using the GraphPad Prism software (v6.0).

## Figures and Tables

**Figure 1 ijms-25-10017-f001:**
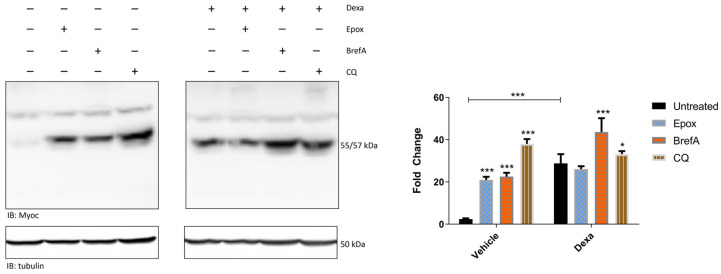
Whole-cell extracts were isolated from TMCs cultivated in the presence or absence of dexa for 4 days and stimulated with epox (1 µM for 3 h), BrefA (20 µM for 2 h) or CQ (20 µM for 2 h) before harvesting, and analyzed by Wb. Filters were probed with antibodies against myocilin (Myoc) and β-tubulin, which was used as loading control. Histograms (right panel) show the fold change of myocilin intensity, normalized against that of tubulin for each experimental condition. Data are expressed as Mean ± SD of *n* = 3 independent experiments. Asterisks refer to the comparison between treated and untreated cells separately, for the two experimental conditions. A comparison between the two untreated conditions was run (indicated by the capped line). One-way ANOVA was followed by Tukey’s post hoc test. * *p* < 0.05, *** *p* < 0.0001.

**Figure 2 ijms-25-10017-f002:**
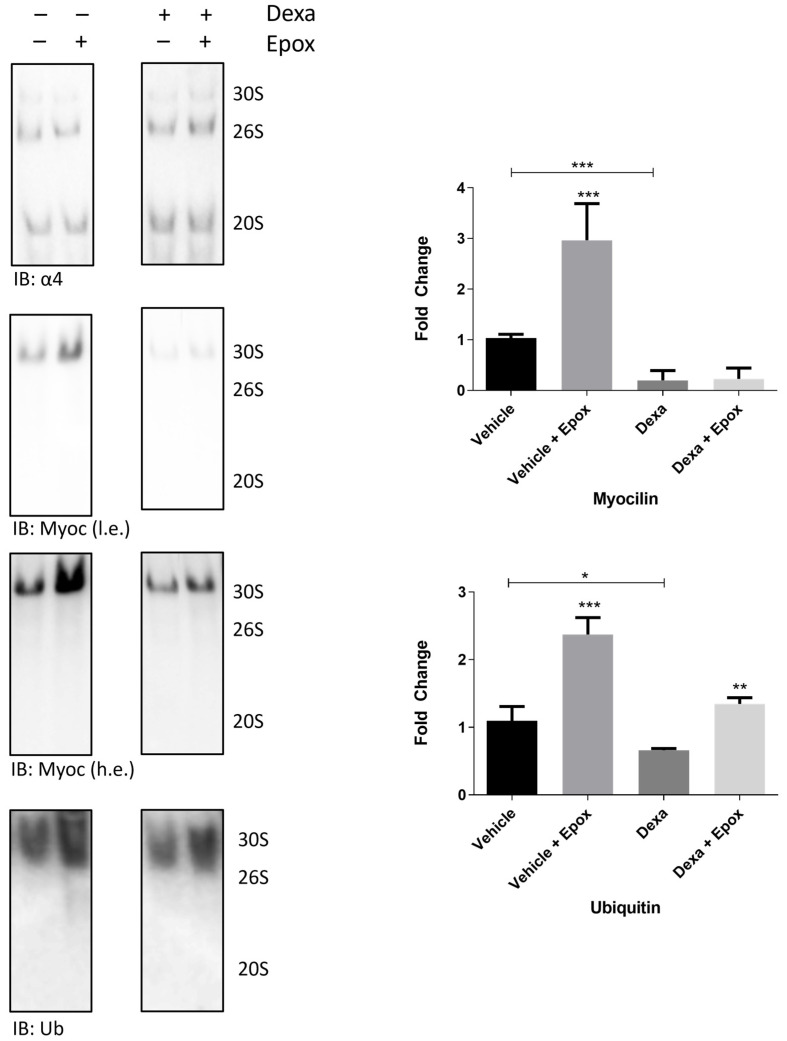
Native proteasome particles were isolated from TMC cultivated in the presence and absence of dexa for 4 days. Epox (1 µM) was delivered 3 h before cell harvesting. Proteasome particles were then transferred to a nitrocellulose filter and probed with antibodies raised against the α4 subunit of the 20S (present in all particles), myocilin (Myoc), shown as low-exposure (l.e.) and high-exposure (h.e.) and ubiquitin (Ub). Histograms (right panel) show the fold change of myocilin and ubiquitin intensity, normalized on that of α4, for each experimental condition. It is worth pointing out that the lanes of myocilin immunostaining (dexa-treated cells) were slightly manipulated, to eliminate an extra band. The original uncropped figure is provided in the “Uncropped Figures” file. Data are expressed as Mean ± SD of *n* = 3 independent experiments. One-way ANOVA was followed by Tukey’s post hoc test. * *p* < 0.05, ** *p*< 0.01,*** *p* < 0.0001.

**Figure 3 ijms-25-10017-f003:**
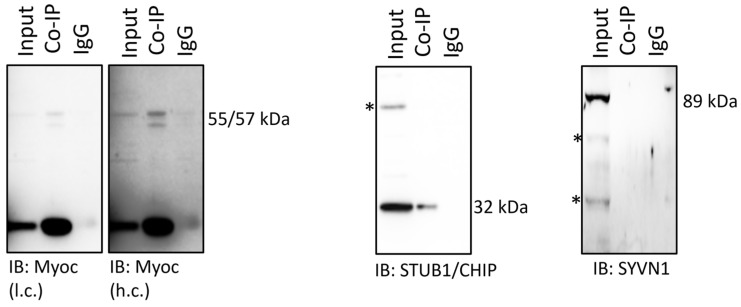
Myocilin was pulled down from whole TMCs lysates (input) using the anti-myocilin antibody. In parallel, non-specific IgGs were used to pull down an equivalent (in terms of protein µg) fraction of the same lysate. The input, together with the IgG pull-down (IgG) and the myocilin pull-down (Co-IP) were analyzed by Wb. Filters were probed with antibodies raised against myocilin (shown as low- and high-contrast exposure), STUB1/CHIP and SYVN1. Asterisks (*) indicate bands of unknown identity.

**Figure 4 ijms-25-10017-f004:**
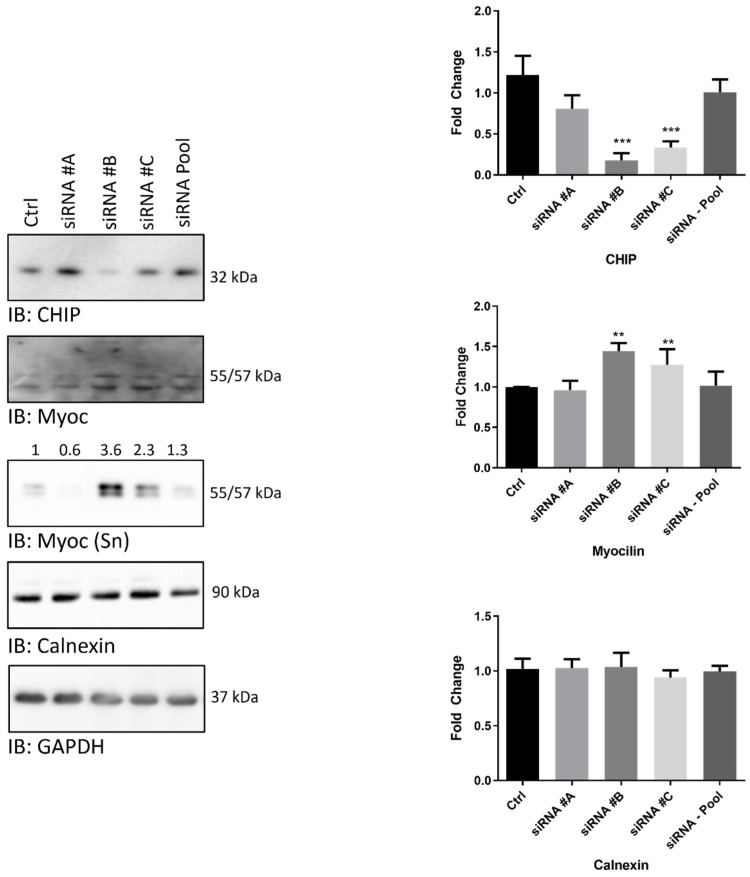
The expression of STUB1/CHIP was silenced by delivery of three different 27-mer siRNAs (#A, #B, and #C) (100 nM, 3-days). As internal control, cells were left untreated or stimulated with a pool of non-targeting siRNA (siRNA–Pool) to rule out off-target effects. Cell lysates were probed with antibodies raised against STUB1/CHIP, Myoc and GAPDH, as loading control. Myocilin in the supernatant fraction of cells was probed and labeled as [IB:Myoc (Sn)] in Figure. In this case, values above the lane represents the calculated variation of Myoc staining adjusted to that of albumin (for details, see Appendix A). In the case of cell lysates, histograms show the band intensity of the indicated protein, normalized against that of GAPDH. Data are expressed as Mean ± SD of *n* = 2 independent experiments. One-way ANOVA was followed by Tukey’s post hoc test. ** *p* < 0.001, *** *p* < 0.0001.

**Figure 5 ijms-25-10017-f005:**
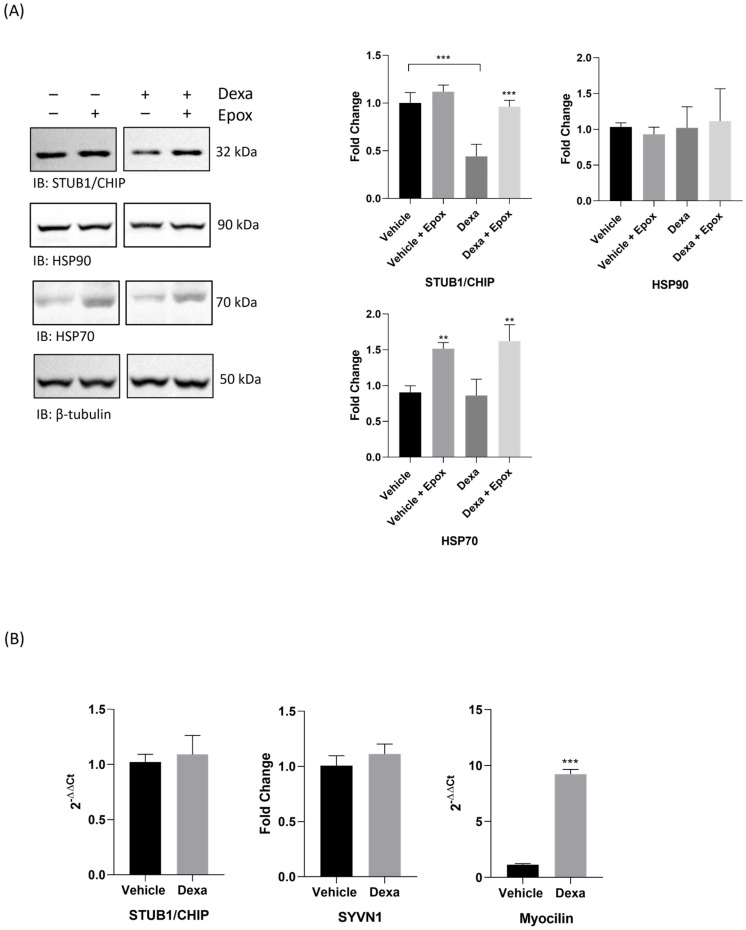
(**A**) Whole-cell extracts were isolated from TMCs cultivated in the presence or absence of dexa for 4 days, and analyzed by Wb. Epox (1 µM) was administered 3 h before cell harvesting. Filters were probed with antibodies raised against STUB1/CHIP, HSP90 and HSP70. β-tubulin was used as loading control. Histograms (bottom panel) show the fold change of STUB1/CHIP, HSP90 and HSP70 normalized on that of β-tubulin, for the different experimental conditions. Data are expressed as Mean ± SD of *n* = 3 independent experiments. One-way ANOVA was followed by Tukey’s post hoc test; (**B**) RT-PCR analysis of STUB1/CHIP, SYVN1 and myocilin transcripts. β-actin was used as a housekeeping gene. Histograms report the fold change of transcript levels normalized on that of actin. Data were calculated using the 2^−ΔΔCt^ formula. Data are expressed as Mean ± SD of *n* = 3 independent experiments. Student’s τ *t* Test, ** *p* < 0.01, *** *p* < 0.0001.

**Figure 6 ijms-25-10017-f006:**
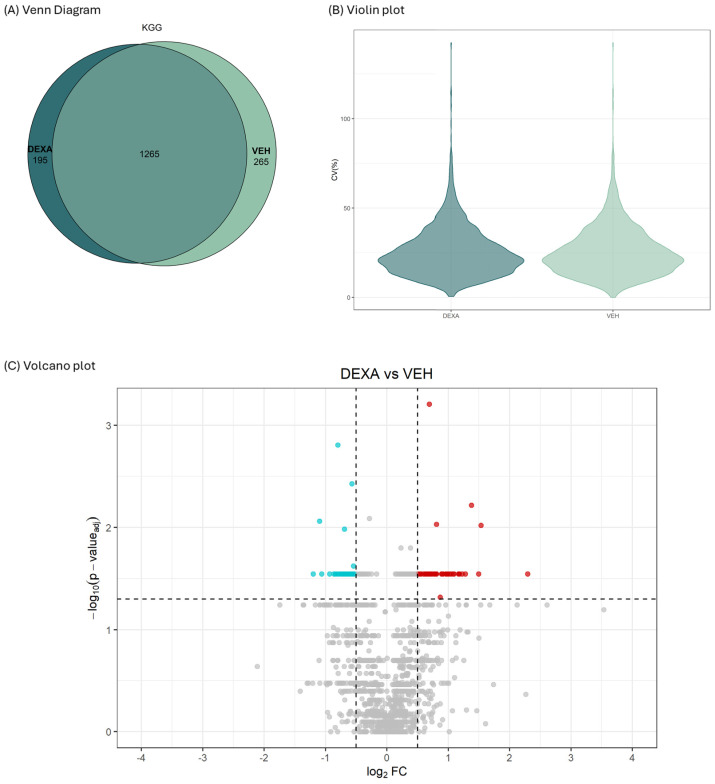
(**A**) Venn diagram showing the fraction of Ub-remnants shared between the two experimental groups and those identified as exclusive; (**B**) violin plot showing the coefficient of variation (CV) of identified and quantified Ub-remnants in dexa- and vehicle (veh)-treated TMCs; (**C**) volcano plot showing Ub-remnants common to dexa- and vehicle-treated cells. A log_2_ fold change (log_2_FC) greater and lower than 0.57 in the dexa vs. vehicle ratio was set as the cut-off. Significance was set as −log_10_ of *p*-value 0.05 (corresponding to 1.3) adjusted by Benjamini–Hochberg correction. Peptides upregulated in dexa-treated cells are in red; peptides downregulated in dexa-treated cells are in turquoise.

**Figure 7 ijms-25-10017-f007:**
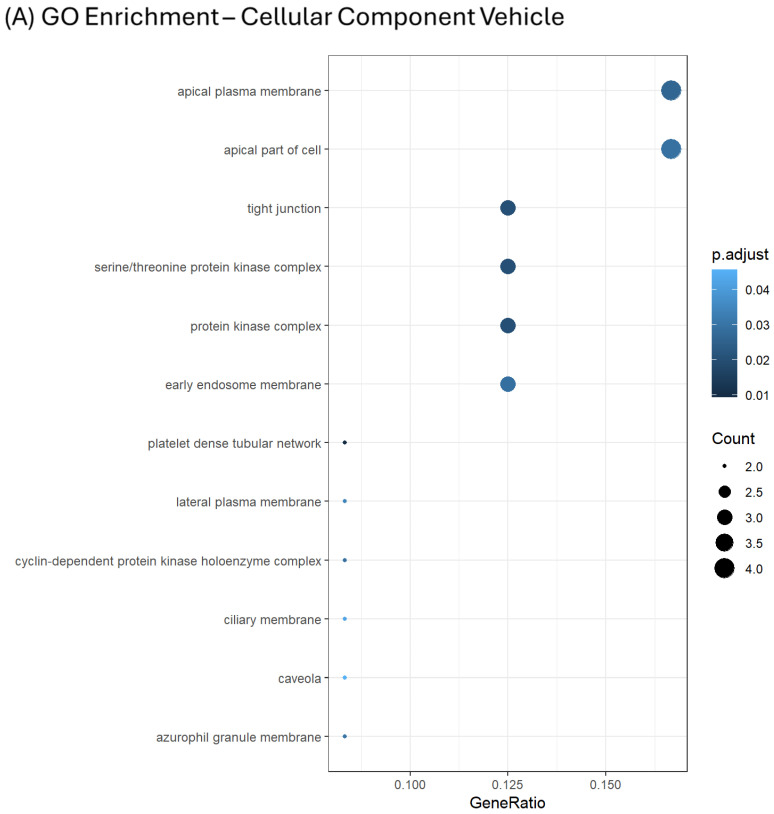
Gene Ontology (GO) enrichment charts of proteins the Ub-peptides were found to belong to. Cellular Component (CC), Molecular Function (MF), and Biological Processes (BPs) were investigated (Vehicle: Panels (**A**,**B**); Dexa: Panels (**C**–**E**)). In the case of vehicle-treated cells, MF retrieved no significantly enriched terms. Count sindicate the number of Ub-proteins associated with the specific GO-term. Cut-off was set for *p* ≤ 0.05 after Benjamini–Hochberg correction.

**Figure 8 ijms-25-10017-f008:**
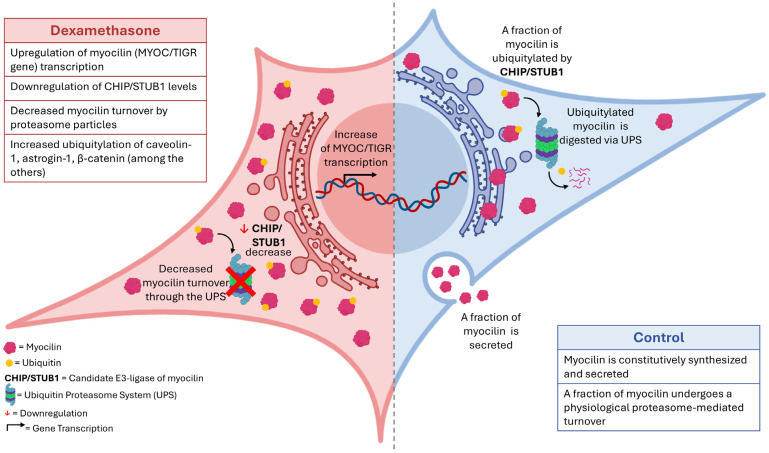
Schematic representation of the main findings of this study. On the left (colored pink), the main alterations induced by dexamethasone treatment are shown. On the right (colored blue), the physiological cycle of myocilin synthesis, turnover and secretion is depicted. The figure was generated with icons from http://BioRender.com.

**Table 1 ijms-25-10017-t001:** List of proteins from which at least one Ub-remnant (diGLY peptide) was generated and found as downregulated in the Dexa/Vehicle ratio. The table shows the protein description and the modified peptide sequence, with the Ub site indicated (as UniMod:121) and the UniProt Accession number.

Downregulated DiGLY Peptides		
Protein Description	Modified-Peptide Sequence	Accession Number
Cell-cycle control protein 50A	(UniMod:1)AMNYNAK(UniMod:121)DEVDGGPPC(UniMod:4)APGGTAK	Q9NV96
G1/S-specific cyclin-D1	AEETC(UniMod:4)APSVSYFK(UniMod:121)C(UniMod:4)VQK	P24385
Matrix remodeling-associated protein 8	AELAHSPLPAK(UniMod:121)YIDLDK	Q9BRK3
Tissue factor	AGVGQSWK(UniMod:121)ENSPLNVS	P13726
G1/S-specific cyclin-D1	AMLK(UniMod:121)AEETC(UniMod:4)APSVSYFK	P24385
Ectopic P-granules protein 5 homolog	AQTQLK(UniMod:121)LPIVPSLQR	Q9HCE0
General transcription and DNA repair factor IIH helicase subunit XPD	C(UniMod:4)QGNLETLQK(UniMod:121)TVLR	P18074
G1/S-specific cyclin-D1	C(UniMod:4)VQK(UniMod:121)EVLPSMR	P24385
Gap junction alpha-1 protein	DC(UniMod:4)GSQK(UniMod:121)YAYFNGC(UniMod:4)SSPTAPLSPMSPPGYK	P17302
Integrin-linked protein kinase	DIVQK(UniMod:121)LLQYK	Q13418
Integrin-linked protein kinase	DTFWK(UniMod:121)GTTR	Q13418
Probable E3 ubiquitin-protein ligase HERC4	ELLDPK(UniMod:121)YGMFR	Q5GLZ8
Probable E3 ubiquitin-protein ligase HERC4	ELVLNGADTAVNK(UniMod:121)QNR	Q5GLZ8
CD320 antigen	ESLLLSEQK(UniMod:121)TSLP	Q9NPF0
Mediator of RNA polymerase II transcription subunit 23	FLSDPK(UniMod:121)TVLSAESEELNR	Q9ULK4
G1/S-specific cyclin-D1	FLSLEPVK(UniMod:121)K	P24385
26S proteasome regulatory subunit 7	FVNLGIEPPK(UniMod:121)GVLLFGPPGTGK	P35998
Glioma pathogenesis-related protein 1	GATC(UniMod:4)SAC(UniMod:4)PNNDK(UniMod:121)C(UniMod:4)LDNLC(UniMod:4)VNR	P48060
Sodium channel protein type 9 subunit alpha	GINYVK(UniMod:121)QTLR	Q15858
Cyclin-dependent kinase inhibitor 1	GLGLPK(UniMod:121)LYLPTGPR	P38936
WD repeat-containing protein 6	GLGVSALC(UniMod:4)FK(UniMod:121)SR	Q9NNW5
Pyruvate kinase PKM	GVNLPGAAVDLPAVSEK(UniMod:121)DIQDLK	P14618
Asparagine--tRNA ligase, cytoplasmic	IFDSEEILAGYK(UniMod:121)R	O43776
Sarcoplasmic/endoplasmic reticulum calcium ATPase 2	IGIFGQDEDVTSK(UniMod:121)AFTGR	P16615
Solute carrier family 12 member 2	K(UniMod:121)ENIIAFEEIIEPYR	P55011
Annexin A1	K(UniMod:121)GTDVNVFNTILTTR	P04083
Myoferlin	KLEPISNDDLLVVEK(UniMod:121)YQR	Q9NZM1
Lysyl oxidase homolog 2	LGQGIGPIHLNEIQC(UniMod:4)TGNEK(UniMod:121)SIIDC(UniMod:4)K	Q9Y4K0
Myoferlin	LIGTATVALK(UniMod:121)DLTGDQSR	Q9NZM1
V-type immunoglobulin domain-containing suppressor of T-cell activation	MDSNIQGIENPGFEASPPAQGIPEAK(UniMod:121)VR	Q9H7M9
Gap junction alpha-1 protein	MGQAGSTISNSHAQPFDFPDDNQNSK(UniMod:121)K	P17302
EH domain-containing protein 1	QLYAQK(UniMod:121)LLPLEEHYR	Q9H4M9
Transmembrane protein 248	QSNPEFC(UniMod:4)PEK(UniMod:121)VALAEA	Q9NWD8
Ephrin type-A receptor 2	QSPEDVYFSK(UniMod:121)SEQLK	P29317
Probable E3 ubiquitin-protein ligase HERC4	SDFFINK(UniMod:121)R	Q5GLZ8
Contactin-associated protein 1	TGTSYFFGGC(UniMod:4)PK(UniMod:121)PASR	P78357
Gap junction alpha-1 protein	YAYFNGC(UniMod:4)SSPTAPLSPMSPPGYK(UniMod:121)LVTGDR	P17302
RNA-binding motif, single-stranded-interacting protein 2	YIK(UniMod:121)TPPGVPAPSDPLLC(UniMod:4)K	Q15434
Golgin subfamily A member 7	YIQEQNEK(UniMod:121)IYAPQGLLLTDPIER	Q7Z5G4
Myoferlin	YTLPLTEGK(UniMod:121)ANVTVLDTQIR	Q9NZM1

**Table 2 ijms-25-10017-t002:** List of proteins from which at least one Ub-remnant (diGLY peptide) was generated and found as upregulated in the Dexa/Vehicle ratio. The table show the protein description and the modified peptide sequence, with the Ub site indicated (as UniMod:121) and the UniProt Accession number.

Upregulated DiGLY Peptides		
Protein Description	Modified-Peptide Sequence	Accession Number
Caveolin-1	(UniMod:1)SGGK(UniMod:121)YVDSEGHLYTVPIR	Q03135
Alpha-enolase	(UniMod:1)SILK(UniMod:121)IHAR	P06733
Cell death-inducing p53-target protein 1	(UniMod:1)SSEPPPPYPGGPTAPLLEEK(UniMod:121)SGAPPTPGR	Q9H305
60S ribosomal protein L24	AITGASLADIMAK(UniMod:121)R	P83731
Phospholipid phosphatase 3	AIVPESK(UniMod:121)NGGSPALNNNPR	O14495
Laminin subunit beta-2	ALESK(UniMod:121)AAQLDGLEAR	P55268
Integrin alpha-5	AQLK(UniMod:121)PPATSDA	P08648
Coiled-coil domain-containing protein 50	AYADSYYYEDGGMK(UniMod:121)PR	Q8IVM0
Annexin A2	DLYDAGVK(UniMod:121)R	P07355
Rho-related BTB domain-containing protein 3	DNYIPVIK(UniMod:121)R	O94955
Myelin protein zero-like protein 1	DYTGC(UniMod:4)STSESLSPVK(UniMod:121)QAPR	O95297
4F2 cell-surface antigen heavy chain	EVELNELEPEK(UniMod:121)QPMNAASGAAMSLAGAEK	P08195
F-box only protein 32	EVYNK(UniMod:121)ENLFNSLNYDVAAK	Q969P5
Store-operated calcium entry-associated regulatory factor	FGK(UniMod:121)TVVSC(UniMod:4)EGYESSEDQYVLR	Q96BY9
F-box only protein 32	FLDEK(UniMod:121)SGSFVSDLSSYC(UniMod:4)NK	Q969P5
Neuroblast differentiation-associated protein AHNAK	FNFSGSK(UniMod:121)VQTPEVDVK	Q09666
Platelet-derived growth factor receptor beta	GDVK(UniMod:121)YADIESSNYMAPYDNYVPSAPER	P09619
Methylsterol monooxygenase 1	IFGTDSQYNAYNEK(UniMod:121)R	Q15800
Dolichyl-diphosphooligosaccharide--protein glycosyltransferase subunit DAD1	IQINPQNK(UniMod:121)ADFQGISPER	P61803
Spartin	IQPEEK(UniMod:121)PVEVSPAVTK	Q8N0X7
Synaptic vesicle membrane protein VAT-1 homolog	ISPK(UniMod:121)GVDIVMDPLGGSDTAK	Q99536
Guanine nucleotide-binding protein G(I)/G(S)/G(T) subunit beta-2	K(UniMod:121)AC(UniMod:4)GDSTLTQITAGLDPVGR	P62879
Actin, aortic smooth muscle	K(UniMod:121)DLYANNVLSGGTTMYPGIADR	P62736; P63267
Actin, cytoplasmic 2	K(UniMod:121)DLYANTVLSGGTTMYPGIADR	P63261
Acyl-CoA (8-3)-desaturase	K(UniMod:121)YMNSLLIGELSPEQPSFEPTK	O60427
Transmembrane protein 45A	LC(UniMod:4)SSEVGLLK(UniMod:121)NAER	Q9NWC5
BCL-6 corepressor-like protein 1	LIVNK(UniMod:121)NAGETLLQR	Q5H9F3; Q6W2J9
Epidermal growth factor receptor substrate 15	LNDPFQPFPGNDSPK(UniMod:121)EK	P42566
Vimentin	LQDEIQNMK(UniMod:121)EEMAR	P08670
Vimentin	LREK(UniMod:121)LQEEMLQR	P08670
Ras-related C3 botulinum toxin substrate 1	LTPITYPQGLAMAK(UniMod:121)EIGAVK	P63000
Potassium voltage-gated channel subfamily E member 4	LWGEAMK(UniMod:121)PLPVVSGLR	Q8WWG9
DNA damage-binding protein 1	MQEVVANLQYDDGSGMK(UniMod:121)R	Q16531
Metalloproteinase inhibitor 1	MYK(UniMod:121)GFQALGDAADIR	P01033
Acyl-CoA (8-3)-desaturase	NK(UniMod:121)ELTDEFR	O60427
Catenin beta-1	NK(UniMod:121)MMVC(UniMod:4)QVGGIEALVR	P35222
Clathrin interactor 1	QDAFANFANFSK(UniMod:121)	Q14677
Prolyl 4-hydroxylase subunit alpha-2	QFFPTDEDEIGAAK(UniMod:121)ALMR	O15460
Calcium-binding and coiled-coil domain-containing protein 2	QNPGLAYGNPYSGIQESSSPSPLSIK(UniMod:121)K	Q13137
4F2 cell-surface antigen heavy chain	QPMNAASGAAMSLAGAEK(UniMod:121)NGLVK	P08195
Vimentin	RQVQSLTC(UniMod:4)EVDALK(UniMod:121)GTNESLER	P08670
Protein HEG homolog 1	SGDFQMSPYAEYPK(UniMod:121)NPR	Q9ULI3
Annexin A2	SLYYYIQQDTK(UniMod:121)GDYQK	P07355
E3 ubiquitin-protein ligase Itchy homolog	SQGQLNEK(UniMod:121)PLPEGWEMR	Q96J02
Potassium voltage-gated channel subfamily E member 4	SSLLLLYK(UniMod:121)DEER	Q8WWG9
60S ribosomal protein L24	TDGK(UniMod:121)VFQFLNAK	P83731
Prolyl endopeptidase FAP	TINIPYPK(UniMod:121)AGAK	Q12884
Vimentin	TNEK(UniMod:121)VELQELNDR	P08670
Procollagen-lysine,2-oxoglutarate 5-dioxygenase 2	VVFAADGILWPDK(UniMod:121)R	O00469
Signal peptidase complex subunit 3	YFFFDDGNGLK(UniMod:121)GNR	P61009
Acyl-CoA (8-3)-desaturase	YMNSLLIGELSPEQPSFEPTK(UniMod:121)NK	O60427
Translocon-associated protein subunit delta	YQVSWSLDHK(UniMod:121)SAHAGTYEVR	P51571

**Table 3 ijms-25-10017-t003:** Lists of primers used for RT-PCR studies.

Gene	5′-3′ Sequence
*β-actin*	GGCCACGGCTGCTTC GTTGGCGTACAGGTCTTTGC
*STUB1*/*CHIP*	AGCAGGGCAATCGTCTGTTC CAAGGCCCGGTTGGTGTAATA
*Synoviolin*/*SYVN1*	CCAGTACCTCACCGTGCTG TCTGAGCTAGGGATGCTGGT
*Myocilin*	AGGTTGGAAAGCAGCAGCC TGCTGTTCTCAGCGTGAGA

## Data Availability

Original uncropped Western blotting figures of horizontally cropped figures are provided in Appendix A. Additional original images, as well as original .raw Orbitrap files of diGLY study and R scripts for analysis of the results, are available on request, by writing to diego.sbardella@fondazionebietti.it.

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
