# Peer review of "The Delayed Turnover of Proteasome Processing of Myocilin upon Dexamethasone Stimulation Introduces the Profiling of Trabecular Meshwork Cells’ Ubiquitylome"

_ijms, 2024, doi:10.3390/ijms251810017_

Round 1

Reviewer 1 Report

Comments and Suggestions for Authors

DELAYED TURNOVER OF PROTEASOME PROCESSING OF MYOCILIN UPON DEXAMETHASONE STIMULATION INTRODUCES THE PROFILING OF TRABECULAR MESHWORK CELLS UBIQUITYLOME

Gel images need to be more clear. 

Glaucoma is chronic optic neuropathy whose pathogenesis has been associated to altered metabolism of Trabecular Meshwork Cells. In this study, the authors have investigated the ubiquitin-mediated turnover of myocilin (MYOC/TIGR ) gene, a glycoprotein with a recognized role in glaucoma pathogenesis, using a human a Trabecular Meshwork strain cultivated in vitro in the presence of dexamethasone. The authors found that dexamethasone treatment downregulates STUB1/CHIP levels by likely promoting its proteasome-mediated turnover. Hence, to strengthen the working hypothesis about global alterations of Ubiquitin-signaling, the first profiling of TMCs ubiquitylome, in the presence and absence of dexamethasone, was here undertaken by diGLY proteomics.

This can help in providing an Ubiquitin centered perspective around the effect of glucocorticoids on metabolism and glaucoma pathogenesis.

Overall the study is well designed and executed. Various sections including the introduction, discussions, conclusion, discussion of results have been presented in a brief and reader-friendly manner. But the quality of the manuscript can be further improved by incorporating the following points into it.

Minor comments:

1. Certaqin gel images needs to be improved. The gels needs to be run again to get a clear crisp bands. For example gels in Fig 2, Fig 3.

2. The title of the article should be in regular sentence mode.

Author Response

Summary

The authors are very grateful to the reviewer for taking the time in evaluating our manuscript and for his/her overall positive comments. We feel that the suggestions have helped us to improve the quality of the manuscript. Thank you.

A point-by-point to the reviewer’s comments is provided below and labeled in italic character.

Point-by-point Comments

Reviewer #1 [Rev. #1]

Gel images need to be more clear. 

Glaucoma is chronic optic neuropathy whose pathogenesis has been associated to altered metabolism of Trabecular Meshwork Cells. In this study, the authors have investigated the ubiquitin-mediated turnover of myocilin (MYOC/TIGR ) gene, a glycoprotein with a recognized role in glaucoma pathogenesis, using a human a Trabecular Meshwork strain cultivated in vitro in the presence of dexamethasone. The authors found that dexamethasone treatment downregulates STUB1/CHIP levels by likely promoting its proteasome-mediated turnover. Hence, to strengthen the working hypothesis about global alterations of Ubiquitin-signaling, the first profiling of TMCs ubiquitylome, in the presence and absence of dexamethasone, was here undertaken by diGLY proteomics.

This can help in providing an Ubiquitin centered perspective around the effect of glucocorticoids on metabolism and glaucoma pathogenesis.

Overall the study is well designed and executed. Various sections including the introduction, discussions, conclusion, discussion of results have been presented in a brief and reader-friendly manner. But the quality of the manuscript can be further improved by incorporating the following points into it.

Minor comments:

  1. Certaqin gel images needs to be improved. The gels needs to be run again to get a clear crisp bands. For example gels in Fig 2, Fig 3.
  2. The title of the article should be in regular sentence mode.

Author’s Reply [A.R.]

Thank you for this criticism. We have edited Figure 2 and Figure 3.

We would like to clarify that, unfortunately, to obtain very crisp bands of proteins bound to native proteasome particles can be very difficult, since these proteins are a very low fraction compared to the proteasome subunits, which, instead, can be successfully immunostained also by setting a low exposure of filters.  

In the case of myocilin-staining we have now replaced the previous Western blot with another one (a low and high exposure filters are proposed) showing a band which reaches a peak of intensity in correspondence of the doubly-capped particle (30S).

In the case of Ub, we have replaced the lanes with two new ones but with very limited improvement of band definition. We apologize for this, but, unfortunately, based on our experience on this project, but also additional ones, it is very difficult to obtain crisp bands using anti-Ub antibodies in native gel electrophoresis. It is our personal opinion that the nature of Ub-proteins and their significant accumulation on proteasome particles cause the Ub-proteins to do not uniformly migrate at least in this assay. However, we hope that the bands now proposed are acceptable at least for a qualitative interpretation of data.    

Regarding the Figure 3 panel (Co-IP study), given the impossibility, at this stage, to run additional co-IP studies, we have tried to improve the quality of the figure presented.

We would like to clarify that detection of the low molecular weight fragment of myocilin (20 kDa) was shown as a further proof of the antibody used for this study. In facts, this fragment is a natural proteolytic product of myocilin processing and can be immunodetected.  Detection of this fragment was shown to compensate for the limited detection of doublet band of myocilin in the case of the Input lane.  

Although it has no impact on data interpretation, since no bands were visible in  both lanes, we clarify that synoviolin staining displayed a wrong lane name attribution (co IP vs IgG).

Reviewer 2 Report

Comments and Suggestions for Authors

This article by Tundo et al. showed that Dexamethasone induced delayed turnover of Myocilin changes the profiling of TMC Ubiquitylome. This is important in the context of disease like Optic neuropathy.

As a reviewer, I have a few questions and suggestions that, if addressed, could enhance the quality of the manuscript:

1.    Please provide a clear biochemical mechanism that supports the paper's conclusions, ideally presented as a comprehensive graphical abstract. This would help improve the article's readability.

2.    Is there any involvement of heat shock proteins, given that this class of proteins may be induced by dexamethasone stimulation? Did the authors investigate this aspect?

3.    The figures should be arranged more effectively. For instance, many of the figures could benefit from separate, labeled panels.
Figure 7, in particular, requires better organization.

Author Response

Summary

The authors are very grateful to the reviewer for taking the time in evaluating our manuscript and for his/her overall positive comments. We feel that the suggestions have helped us to improve the quality of the manuscript. Thank you.

A point-by-point to the reviewer’s comments is provided below and labeled in italic character.

Point-by-point Reply

Reviewer #2 [Rev. #2]

This article by Tundo et al. showed that Dexamethasone induced delayed turnover of Myocilin changes the profiling of TMC Ubiquitylome. This is important in the context of disease like Optic neuropathy.

As a reviewer, I have a few questions and suggestions that, if addressed, could enhance the quality of the manuscript:

  1. Please provide a clear biochemical mechanism that supports the paper's conclusions, ideally presented as a comprehensive graphical abstract. This would help improve the article's readability.

Author’s Reply [A. R.]

We agree that a comprehensive graphical abstract would have helped the readers to follow our paper, thank you. We have now drawn and incorporated (Figure 8) a representation of our main findings. 

  1. Is there any involvement of heat shock proteins, given that this class of proteins may be induced by dexamethasone stimulation? Did the authors investigate this aspect?

[A.R.]

We are very grateful to the reviewer for this comment, thank you. We did not consider this possibility. We have now assayed the content of two major Heat Shock Proteins (HSPs), such as HSP90 and HSP70, which were both proposed to interact and modulate STUB1/CHIP activity.

Moreover, we reasoned that to assay the content of HSP90 and HSP70 in the presence/absence of epoxomicin would have been interesting to figure out whether the behavior observed for STUB1/CHIP in the presence of dexamethasone stimulation and proteasome inhibition was specific for this last protein or not. Therefore, we measured the content of the HSPs under these experimental conditions.

As discussed in Figure 5, we observed that the basal content of HSP70 and HSP90 was unaltered by dexa treatment alone, compared to control cells. In the presence of epoxomicin, HSP90 levels were unchanged in all cases, whereas in the case of HSP70, a significant accumulation of the protein was observed, at comparable rates, for both dexa- and vehicle-treated cells. 

These data are shown and discussed in the Results and Discussion sections.

  1. The figures should be arranged more effectively. For instance, many of the figures could benefit from separate, labeled panels.
    Figure 7, in particular, requires better organization.

[A. R.]

Thank you for this criticism. We agree that some of the figures could have been presented in a more clear fashion.

In particular, we have now split the panels of Figure 5 to better distinguish Wb from RT-PCR data and we have edited the panel of diGLY data, in particular the GO enrichment charts further providing a new label for each one of the five tables.

We hope we interpreted correctly the reviewer’s criticism and that in this edited version of the manuscript the figures are more clear. Of course, the authors are available to further edit the figures based on additional comments.  

Round 2

Reviewer 2 Report

Comments and Suggestions for Authors

I am overall happy with the changes that were made after a thorough revision. 

A minor issue that should be taken care of, is the figure fonts are very small (For eg. Fig 5) and not consistent .

Thanks for addressing the comments.